# Influence of Floodplain Flooding on Channel Siltation Adjustment under the Effect of Vegetation on a Meandering Riverine Beach

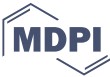

**Junhua Li [1], Mingwu Zhang [1], Enhui Jiang [1,\*], Li Pan [1], Aoxue Wang [2], Yafei Wang [2] and Shengqi Jian [2]**

1   Yellow River Institute of Hydraulic Research, Yellow River Conservancy Commission, Zhengzhou 450003, China; ljhyym@126.com (J.L.); thuzmw08@126.com (M.Z.); blondepan@126.com (L.P.)
2   College of Water Conservancy Science and Engineering, Zhengzhou University, Zhengzhou 450001, China; wangaoxuer@163.com (A.W.); wangyafeizzu@163.com (Y.W.); jiansq@zzu.edu.cn (S.J.)
\*   Correspondence: hkyjeh@163.com

**Abstract:** Flooding in a sediment-laden floodplain is affected by riverine beach vegetation and the shape of a meandering compound channel. The laws of water and sediment exchange and the deposition distribution in beach troughs are very complex. These factors play a significant role in the formation and development of secondary suspended rivers, in the adjustment of the beach horizontal gradient, and even in the evolution of the flood control situation. In this study, we used a combination of experimental simulation and theoretical research to carry out a generalized model test of floodplain flooding evolution, analyzed the lateral distribution characteristics of sediment-laden flow and sediment factors in a meandering compound channel under the conditions of beach vegetation, and revealed the pros and cons of beach vegetation on the adjustment of the beach and channel siltation. The model test results of the flooding in the floodplain in the compound channel with meandering vegetation showed that the main stream was not only concentrated in the main channel but also appeared near the foot of the left and right bank levees and formed flood discharges along the embankment. As the riverine beach siltation was mainly concentrated at the riverine beach lip, the vegetation on the riverine beach had a significant effect on slowing down the flow velocity. Whether it was a row or full vegetation on both sides of the bank, this played an important role in the stability of the main channel. When there was no vegetation on the riverine beach, the main channel was easy to move. The arrangement of full vegetation on the riverine beach had a uniform effect on the velocity distribution of the riverine beach, which reduced the phenomenon of excessive velocity at the foot of the riverine beach and increased the velocity effect in the main channel. These results will provide a theoretical basis for the utilization of riverine beach areas and river management in the lower Yellow River and have a great significance for enriching the basic theory of water and sediment movement and promoting the integration of hydraulics, river dynamics, and ecology.

**Keywords:** curve riverine beach vegetation; floodplain flood; law of water and sediment; riverine beach groove deposition; model experiment

## 1. Introduction

Some rivers in the natural world (such as the middle and lower reaches of the Yellow River) are open and flat, and the ability of the river to carry solid materials is low. As a result, a large amount of bed loads and suspended loads are deposited along the river every year, and the riverbed is raised year by year. There are many banks and forked trenches, which alternate with riverine beaches and bays. The river channel is meandering, and the riverbed swings from left to right [1–4]. The riverbed is uneven in width and very shallow, and the flood and dry flow varies greatly. The river easily forms wide, shallow, scattered, and meandering typical plains and small flood plains, which bring new difficulties to flood control. For this type of river, people began to explore the use of compound cross-sections to remediate the river [5–8].

The sediment-laden floodplains are affected by riverine beach vegetation and the shape of the meandering compound channel. The laws of water and sediment exchange and the deposition distribution in riverine beach trough are complex and play a significant role in the formation and development of a secondary suspended river (The suspended secondary river is relative to the first class suspended river of the Yellow River. The bottom of the main channel is higher than the middle bank.), the adjustment of the riverine beach horizontal gradient, and even the evolution of the flood control situation [9]. The aspect of flood plain water and sediment structure under the action of vegetation in a meandering compound channel riverine beach is mainly seen in the research of clear water flow and bedload movement law. However, for sandy rivers, sediment laden flow is one of the characteristics that cannot be described by the existing clear water flow and bedload movement rules in terms of the physical characteristics, movement characteristics, and sediment transport characteristics.

It is necessary to improve the existing theoretical model and to establish a lateral distribution model of water and sediment transport under the coupling effect of meandering complex channels and riverine beach vegetation that is suitable for sediment laden floodplain flow [10–12]. Due to the addition of the riverine beach vegetation and meandering compound river channel, it is difficult to study the influence of floods in sediment-laden floodplains on riverbed shape adjustment. The coupling of any two factors is complicated, including the bend, compound channel, riverine beach vegetation, sediment-laden flow, and so on. How to consider the impact of sandy floodplains on the adjustment of riverbed morphology under the coupling effect of a meandering compound channel and riverine beach vegetation is an unresolved scientific problem in the study of river dynamics and riverbed evolution [13–17].

In the past, little attention has been paid to the influence of forests and high stalk crops on the movement and deposition of riverine beach sediment in the study of the water and sediment exchange in the lower Yellow River. River flow with vegetation is a special and complex flow problem [17,18]. The traditional resistance equations (such as the Manning, Chezy, and Darcy–Weisbach equations) used in the hydraulic calculation of river flow with vegetation will produce large errors. The main reason is that the resistance of vegetation to flow is the vegetation's water blocking effect rather than the bed resistance [18–22].

This is shown as follows: (1) the vegetation itself, including its shape, density, whether it is submerged, and its stiffness or flexibility; (2) regarding the vegetation and the water flow from the perspective of hydrodynamics, each plant is actually a very complex circular cylinder flow problem; and (3) there is a strong interaction between the vegetation area and non-vegetation area water flow as well as a strong momentum exchange between the flow of the riverine beach channel. A strong turbulent vortex street is formed near the riverine beach and trough interface, which is more prominent and complex when there is vegetation on the riverine beach, and the existence of the vortex also hinders the smooth flow of water [23–25].

In the lower Yellow River, when the sediment laden flood occurs, this is intertwined with the influence of the riverine beach vegetation and the meandering channel, resulting in a complex distribution and flow structure of the water and sediment, which directly affects the flow exchange of the riverine beach and channel, the distribution of the riverine beach sedimentation, and the effect of the scouring channel. This is an important reason for the secondary suspended river and the horizontal slope of the riverine beach surface [26,27]. Therefore, research on the effects of meandering compound channels and riverine beach vegetation on sand-bearing flow movement and riverine beach channel siltation has an important practical significance for the management of secondary suspended rivers in the lower Yellow River.

In this study, we used a combination of experimental simulation, actual measurement data analysis, and theoretical research to establish a meandering compound river model of riverine beach vegetation, then conducted a generalized model test for the evolution of the suspended load floodplain, and analyzed the transverse distribution characteristics of

the flow and sediment factors in a meandering compound channel under the influence of riverine beach vegetation, and revealed the pros and cons of floods on the riverine beach of floodplains and the channel siltation adjustment under the condition of beach vegetation. This achievement will provide a theoretical basis for riverine beach area applications and river regulation in the lower Yellow River and have great significance in enriching the basic theory of water and sediment movement and promoting the integration of hydraulics, river dynamics, and ecology.

## 2. Materials and Methods

### 2.1. Test Setup

2.1.1. Model Design

Based on the practical consideration of the form of meandering river in the lower reaches of the Yellow River, we established a meandering compound channel model. The test used an electric water pump to pump water from an underground reservoir into a tank, and the flow of clean water was controlled by an electromagnetic flowmeter and flow control system. The flow of muddy water was controlled by an orifice box. The tank was 60 m long, 7 m wide, and 0.7 m high (Figure 1). The walls on both sides of the sink were made of brick and cement walls, and the bed surface gradient was 0.2%. To control the water level in the tank, the tail of the tank was equipped with an adjustable electric tail gate. The water flowing from the tail of the tank flowed into the underground reservoir. The total width of the meandering compound channel section was 7 m, and the width of main channel was 70 cm.

The inner radius of the 120° arc of the bending section was 150 cm, and the outer radius was 220 cm; the adjacent arc sections were connected by straight lines. The shortest distance between the top of the meandering arc on both banks was 150 cm. In the cross section of an arc top section of a meandering compound channel, the width of the floodplain on the left bank was 150 cm, the width of the floodplain on the right bank was 480 cm, and the height of the floodplain vegetation was 7 cm, and the depth was 7 cm (Figure 2).

2.1.2. Measuring Equipment

An LS300-A portable current meter produced by Nanjing Zhuoma electromechanical Co., Ltd. (Nanjing, China) was used in this test, and the pycnometer method was used to measure the sediment concentration. The measuring instruments included a pycnometer and high-precision balance, and the particle size of the sediment was measured by laser particle size analyzer. The median particle size is 0.3 mm. The particle size distribution presented a single peak, and the sediment uniformity was relatively high. D03, D06, D10, D16, D25, D50, D75, D84, D90, and D97 were 0.153, 0.170, 0.186, 0.205, 0.229, 0.292, 0.378, 0.427, 0.473, and 0.573 mm, respectively.

2.1.3. Vegetation Arrangement

Disposable chopsticks (22.5 cm in length and 5 mm in diameter) were glued on the riverine beach to simulate vegetation. The height and diameter of the floodplain vegetation were 7 cm and 5 mm, respectively. The experiment considered the vegetation arrangement in five cases: (1) a meandering compound channel with no vegetation on the riverine beach; (2) a meandering compound channel with vegetation on the convex bank of the riverine beach; (3) a meandering compound channel with vegetation on the concave bank of the riverine beach; (4) a meandering compound channel with vegetation on both sides of the beach; and (5) a meandering compound channel with full vegetation on the beach. Five kinds of vegetation were arranged in the water tank from the upper section to the lower section, each with a length of 8 m and two arc bends, one on the left and one on the right bank (Figure 1).

Based on practical considerations, vegetation increased gradually from upstream to downstream. Therefore, our model gradually increases vegetation from upstream to downstream (from CS1–CS29).

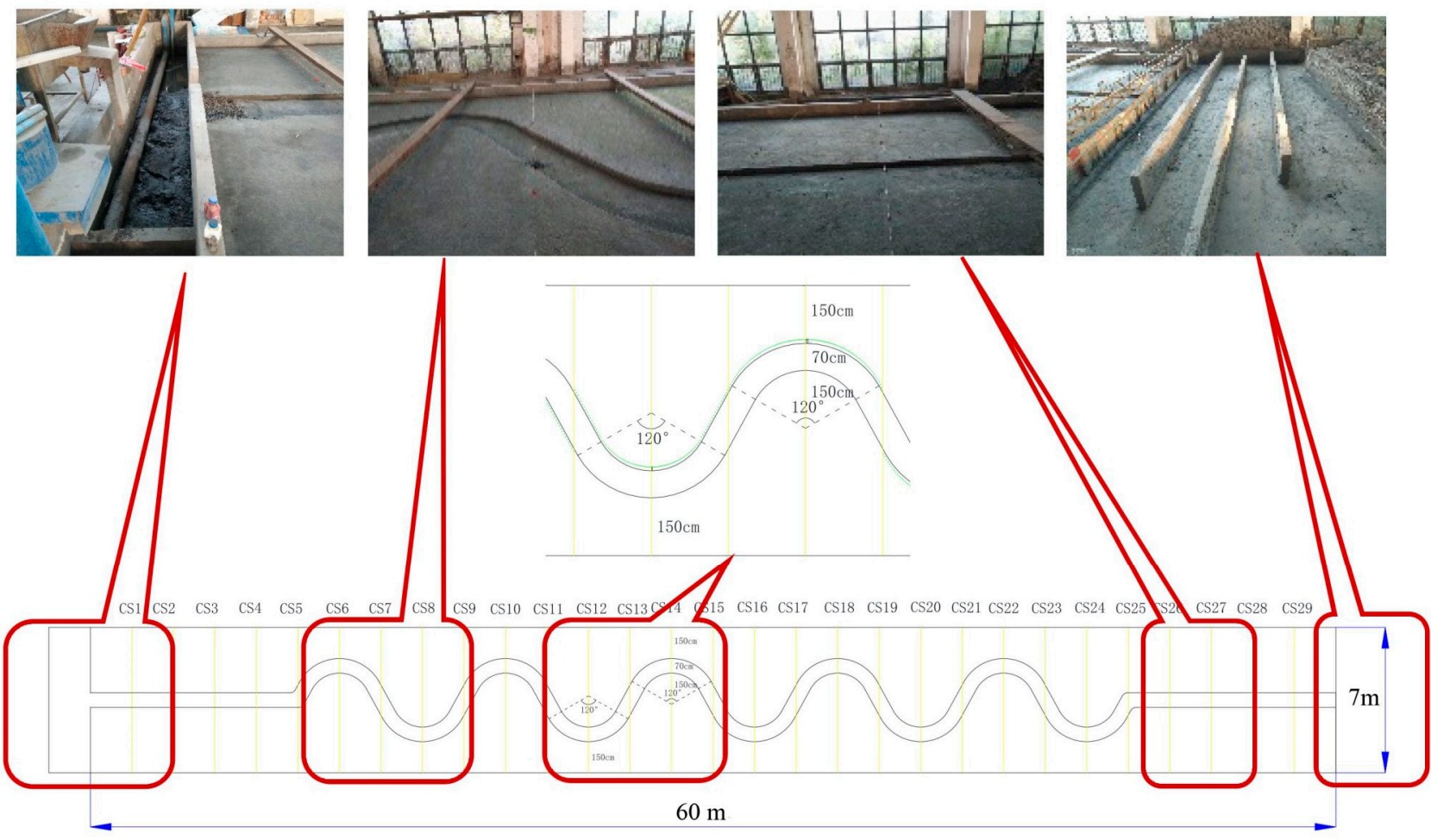

**Figure 1.** Layout of the generalized flume model.

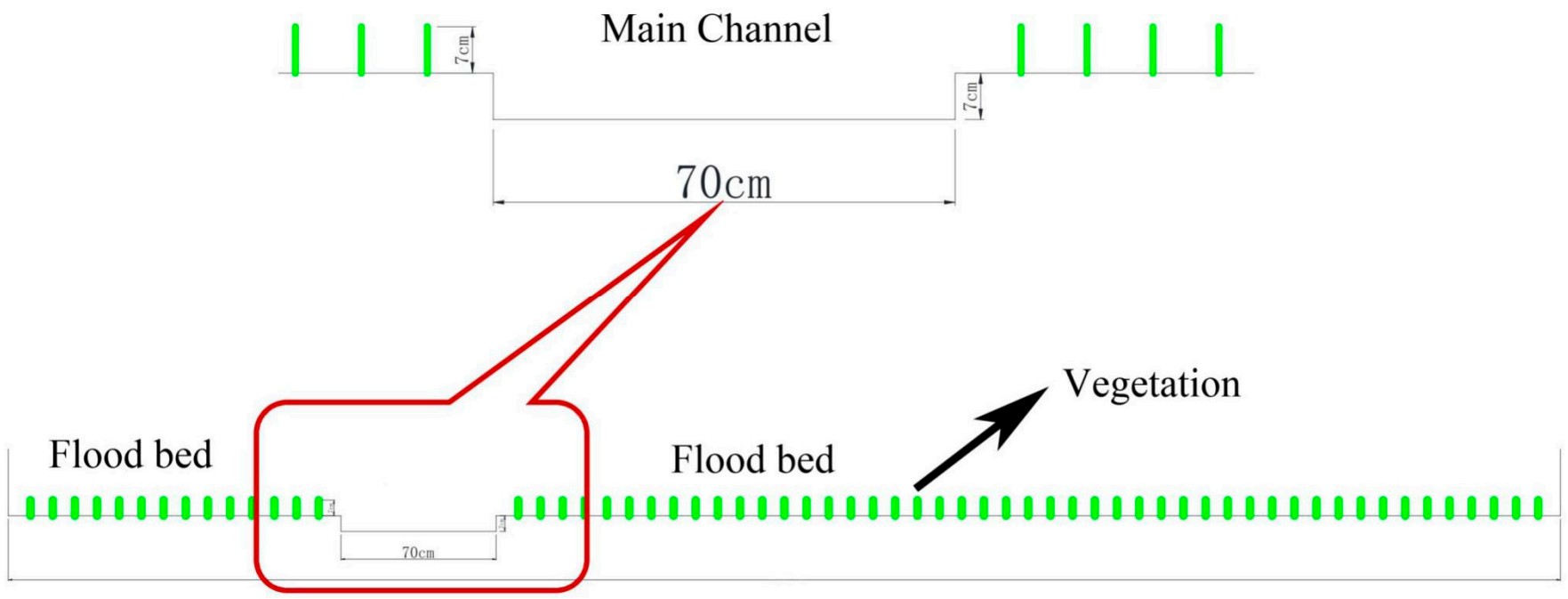

**Figure 2.** Cross section of the arc top section of the meandering compound channel.

### 2.1.4. Layout of Measuring Points

The topographic survey sections: CS1–CS29. At the end of each condition test, the shape of each section was measured.

The flow velocity measurement sections: CS6, CS8, CS10, CS12, CS14, CS16, CS18, CS20, CS22, and CS24; the measuring point layout of the flow velocity measurement section: one measuring point every 0.2 m. During the test process of each condition, the velocity of the above section was monitored in real time.

The sand content measurement sections: CS3, CS8, CS12, CS16, CS20, CS24, and CS27. During the test process of each condition, the sand content measurement of the above section was monitored in real time (Figure 1).

The connection between the inlet channel and the meandering channel is a very abrupt change, which influences the behavior of the cross section of CS6 and CS8. However, the influences are limited, this is a transition zone, so we did not set up vegetation in the cross section of CS6 and CS8.

### 2.1.5. Test Conditions

To study the flow and sediment situation of a meandering compound channel with vegetation on the riverine beach more comprehensively, seven kinds of working conditions were designed through different combinations of sediment concentrations and sediment particle size. The design flow was 100 m$^3$/h, and the actual situation was a little different.

Condition 0 was the initial condition to adapt to the flow conditions of the initial design terrain; condition 1 was under the condition of clear water; conditions 2–4 were under the conditions of relatively fine sediment to consider different sediment concentrations; and conditions 5–7 were under the condition of relatively coarse sediment to consider different sediment concentrations. The different sediment concentrations were about 5 kg/m$^3$ for small sediment concentration, 14.5 kg/m$^3$ for medium sediment concentration, and 35.3 kg/m$^3$ for large sediment concentration (Table 1). The test tailgate could automatically adjust the height and control the water level of the tailgate. The Reynolds number in CS24 near downstream tailgate is about 5388, and the Froude is about 0.051. The numbers in other sections are greater than these numbers.

**Table 1.** Test conditions.

| Working Condition | Design Flow (m$^3$/h) | Actual Flow | Design Sediment Concentration | Actual Sediment Concentration (kg/m$^3$) | Design Particle Size |
|:---:|:---:|:---:|:---:|:---:|:---:|
| 0 | 100 | 100.2 | no | 0 | no |
| 1 | 100 | 90.7 | no | 0 | no |
| 2 | 100 | 93.1 | small | 5.23 | fine |
| 3 | 100 | 101.3 | large | 35.37 | fine |
| 4 | 100 | 104.8 | middle | 14.39 | fine |
| 5 | 100 | 100.6 | small | 4.84 | coarse |
| 6 | 100 | 101.5 | middle | 14.85 | coarse |
| 7 | 100 | 101.5 | large | 35.30 | coarse |

The tests were carried out in working conditions 0 to 7 successively, and the section was measured after the end of the tests in each working condition. Restricted by the test conditions, the test duration of each working condition was 1.5 h. The equilibrium is difficult to achieve because the terrain is constantly changing, even at condition 7.

### 2.2. Effect of Bend on Transverse Distribution of Floodplain Flow

In this study, the Shiono and Knight method (SKM) [28–30] was used to analyze the force acting on the flow micro control body. Based on the momentum equation considering the lateral secondary flow inertia force and channel curvature, a lateral velocity distribution model of the composite section was established to reveal the action mechanism of different curvature boundary conditions on the transverse distribution of the flood and sediment.

Although SKM was developed for straight compound channels, it has been used for meandering compound channels and there have been some relevant studies [15,29,30].

### 2.2.1. Governing Equation of Water Flow

The force analysis of the flow micro control body was carried out. Based on the momentum equation considering the inertial force of lateral secondary flow and the curvature of the river channel, the transverse velocity distribution model of the compound section was established. For the meandering compound rectangular channel, the meandering top section is shown in Figure 3, and the vertical average velocity and transverse distribution of the sediment carrying capacity were calculated. The meandering top section of the meandering compound rectangular channel included B1 = 20 cm, b = 20 cm, B2 = 80 cm, h = 20 cm, H = 49 cm, and S0 = 0.001. The different r values were 20, 40, 60, and 80 cm, respectively.

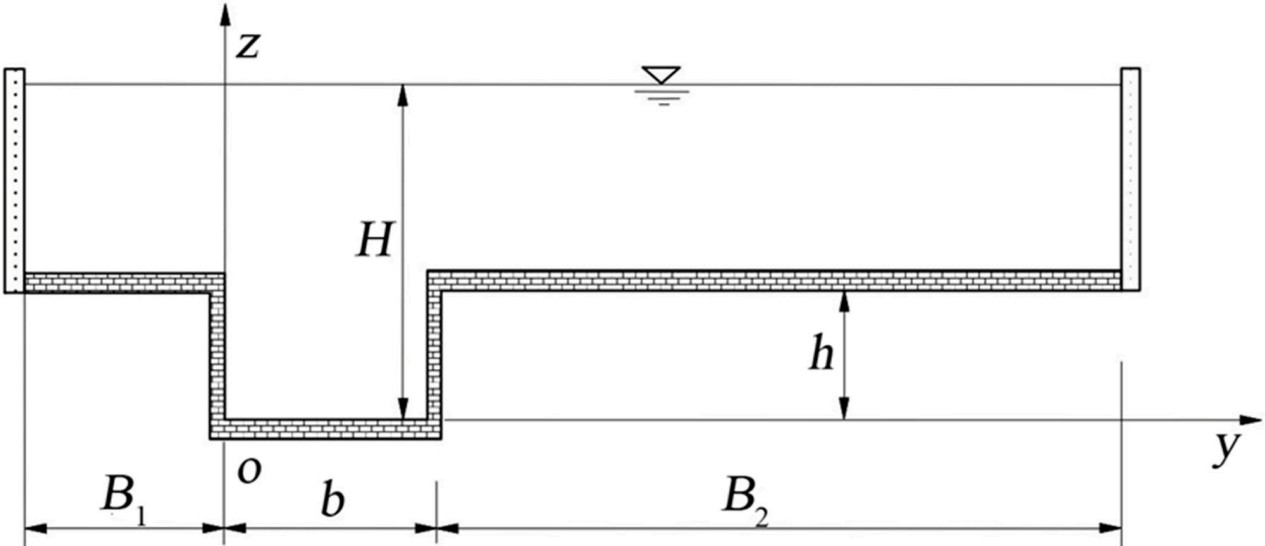

**Figure 3.** Schematic diagram of the meandering arc top section of a meandering compound rectangular channel.

In order to calculate, first of all, the governing equations for the left riverine beach, the middle main channel and the right riverine beach were given as follows:

$$U_d^{(1)} = \left(A_1 e^{\gamma_1 y} + A_2 e^{-\gamma_1 y} + k_1\right)^{1/2} \tag{1}$$

$$U_d^{(2)} = \left(\frac{8r}{r+y}\frac{\rho g H S_0 - \Gamma_{mc}}{\rho f}\right)^{1/2} \tag{2}$$

$$U_d^{(3)} = \left(A_3 e^{\gamma_3 y} + A_4 e^{-\gamma_3 y} + k_3\right)^{1/2} \tag{3}$$

Among them:

$$k_1 = \frac{8g(H-h)S_0}{f_1}\left(1 - \frac{\Gamma_1}{\rho g(H-h)S_0}\right) \tag{4}$$

$$\gamma_1 = \sqrt{\frac{2}{\lambda_1}}\left(\frac{f_1}{8}\right)^{1/4}\frac{1}{H-h} \tag{5}$$

$$k_3 = \frac{8g(H-h)S_0}{f_3}\left(1 - \frac{\Gamma_3}{\rho g(H-h)S_0}\right) \tag{6}$$

$$\gamma_3 = \sqrt{\frac{2}{\lambda_3}}\left(\frac{f_3}{8}\right)^{1/4}\frac{1}{H-h} \tag{7}$$

In the formula, $U_d$ is the vertical average velocity in the downstream direction; $\rho$ is the fluid density; $g$ is the acceleration of gravity; $H$ is the water depth; $S_0$ is the bed slope in the downstream direction; $f$ is the bed friction factor; $r$ is the available curvature of bed slope in the main channel and riverine beach; y is the transverse coordinate; $\Gamma$ is the quadratic flow term; $\gamma$ is the exponent of the analytical solution of the differential equation; and $A$ is the unknown constant.

### 2.2.2. Boundary Conditions

The advantage of the SKM is that it can be solved analytically. In order to solve the control equation given above, it was necessary to determine the boundary conditions.

$$U_d^{(1)}\Big|_{y=-B_1} = 0 \; ; \; U_d^{(1)}\Big|_{y=0} = U_d^{(2)}\Big|_{y=0} \; ; \; U_d^{(2)}\Big|_{y=b} = U_d^{(3)}\Big|_{y=b} \; ; \; U_d^{(3)}\Big|_{y=b+B_2} = 0 \quad (8)$$

By using four boundary conditions to substitute for Equations (1) and (3), the values of the coefficients $A_1$, $A_2$, $A_3$, and $A_4$ can be obtained by solving this linear equation system. The transverse distribution of the average velocity of the vertical line of the meandering compound rectangular compound channel was obtained.

## 3. Results and Discussions

### 3.1. Characteristics of the Water and Sediment Transport of a Flood under the Effect of Vegetation in a Bend Floodplain

(1) A meandering compound channel with no vegetation on the riverine beach (CS6 and CS8). With the experimental conditions, the channel did not have a balanced deposition, and the velocity distribution changed from relatively uniform (the velocity in the main channel was relatively large) to extreme (the velocity was not only concentrated in the main channel but also in the riverine beach wall). Due to the bending of the main channel in the initial topography, the main channel deviated to the left or right bank. With the siltation evolution of the channel, the main channel moved to the middle. From the velocity chart, the velocity of the convex bank was greater than that of the concave bank, and there was a trend of large flood straightening (Figure 4).

(2) A meandering compound channel with vegetation on the convex bank of the riverine beach (CS10 and CS12). Similar to the meandering compound channel without vegetation on the riverine beach, the channel silted up unevenly, and the velocity distribution changed from relatively uniform (the velocity in the main channel was relatively large) to extreme (the velocity was not only concentrated in the main channel but also on the side wall of the riverine beach); however, the difference was that the main channel of the meandering compound channel with vegetation on the convex bank of the riverine beach essentially had no moving position.

The reasons were as follows: first, the vegetation on the convex bank played a role to reduce the flow velocity and slow down or prevent the erosion of the vegetation on the convex bank; secondly, the meandering compound channel with vegetation on the convex bank of the riverine beach was downstream of the meandering compound channel without vegetation on the riverine beach, and the erosion ability of the main channel from upstream to downstream was weakened (Figure 4).

(3) A meandering compound channel with vegetation on the concave bank of the riverine beach (CS14 and CS16). As the meandering compound channel with vegetation on the concave bank of the riverine beach was downstream of the meandering compound channel with vegetation on the convex bank of the riverine beach, the flood flow rate of the large floodplain was relatively uniform. Second, as the main flood channel was meandering, and the flow rate was lower than that on the riverine beach, and the flow velocity of the main channel could appear less than the flow rate of the riverine beach. Third, with the formation of upstream floods (especially at the left bank side wall), the side wall had no vegetation to form a new flow channel and continued to flow downstream (Figure 4).

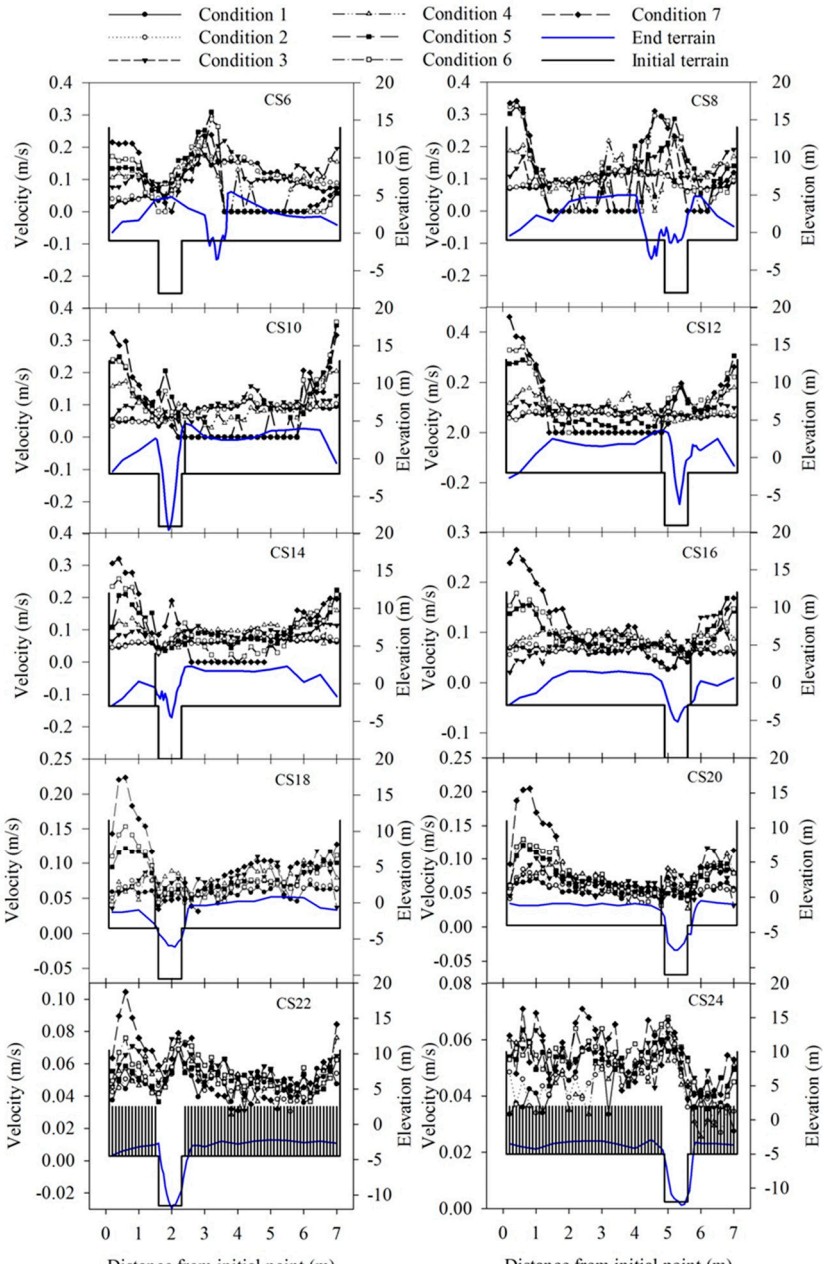

**Figure 4.** Comparison of the velocity and topography of meandering compound channel sections under different working conditions (The blue line is the terrain at the end of all working conditions).

(4) A meandering compound channel with vegetation on both sides of the riverine beach (CS18 and CS20). The siltation of the river channel was relatively uniform. Excluding the flood along the dike formed upstream (especially the left bank side wall), there was no vegetation on the side wall that formed a new channel for water flow, which continued to flow downstream, and the velocity distribution was relatively uniform. There was a row of vegetation on each side of the bank, the vegetation had an obvious water blocking effect, and the velocity of the vegetation was relatively small (Figure 4).

(5) A meandering compound channel full of vegetation on the riverine beach (CS22 and CS24). The siltation of the river channel was relatively uniform. The upstream formed floods along the dike (especially at the left bank side wall), due to the side wall vegetation, and the water flow was subject to resistance, which made the riverine beach velocities relatively uniform. The flow velocity of the main channel on the convex bank was larger

than that on the concave bank, which had the characteristics of a large floodplain with a meandering compound channel (Figure 4).

### 3.2. Effect of the Bend Curvature on the Transverse Distribution of the Flow and Sediment Factors in Compound Channels

The larger the curvature radius r (the smaller the curvature of the channel), the flatter the velocity distribution, and the smaller the difference of the vertical average velocity between the left and right banks of the main channel (Figure 5A). After obtaining the transverse velocity distribution of the floodplain flow in the meandering compound channel, the sediment carrying capacity formula was further introduced. Here, the Zhang Hongwu formula [31] was used to calculate the vertical average sediment carrying capacity transverse distribution of the floodplain flow in a meandering compound channel.

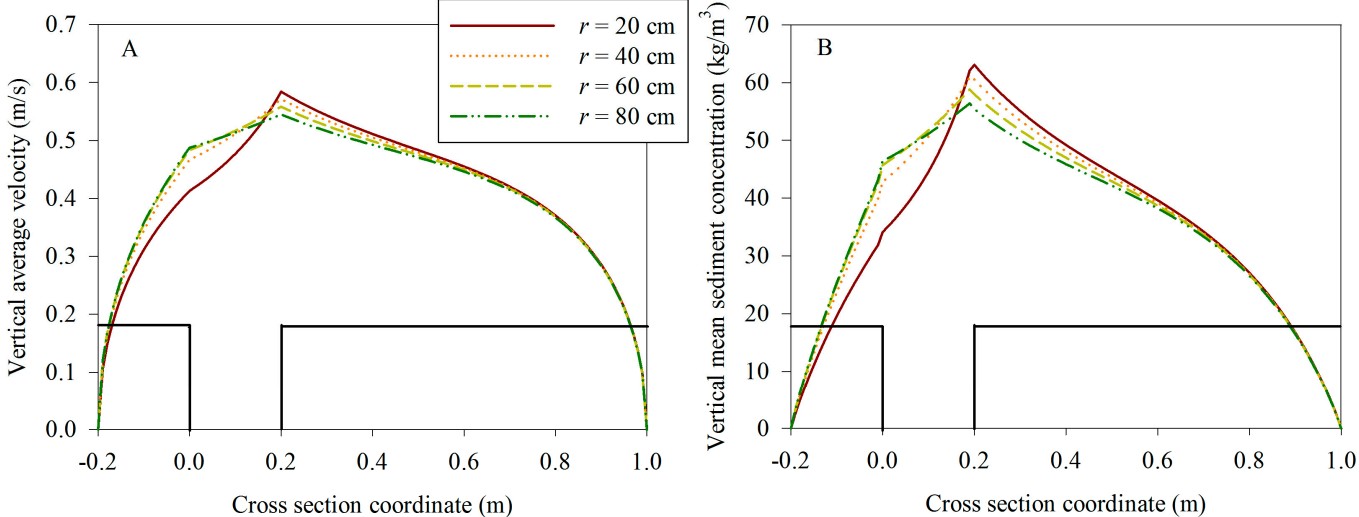

**Figure 5.** Transverse distribution of the top flow, sediment carrying capacity, and velocity in a meandering compound rectangular channel. (**A**), effect of the bend curvature on the transverse distribution of the flow; (**B**) effect of the bend curvature on the transverse distribution of the sediment carrying capacity.

The transverse distribution of the vertical average sediment carrying capacity under different curvature conditions (Figure 5B) was obtained. When the curvature radius r was larger (the curvature of the channel was smaller), the distribution of the sediment carrying capacity was smoother, and the difference of the vertical average sediment carrying capacity between the left and right banks of the main channel was smaller. Therefore, the transverse distribution of the flow and sediment factors was different with different curvatures of the bend. The influence of the curvature of the bend on the transverse distribution of the flow and sediment factors was also demonstrated.

### 3.3. Effect of Floodplain Flooding on the Channel Filtation Adjustment under the Effect of Riverine Beach Vegetation

As there were many test scenarios, particular sections were selected for specific analysis:

(1) Adjustment of the cross-section shape of a straight compound river channel without vegetation on the front riverine beach (CS1–CS5). At section 1, the riverine beach was gradually silted and heightened as the test progressed; and, after working condition 1 (clear water) and working condition 2 (small sand content), the main trough was scoured, and the elevation of the deep point was increased from −13 to −17.5 cm, after which the main trough was gradually silted and raised, and the elevation of the deep point was raised to −4.2 cm. The position of the main channel was relatively unchanged after the first three working conditions, and the position of the main channel was gradually shifted to the left after the fourth working condition.

At section 5, the riverine beach was gradually silted and heightened as the test progressed. The riverine beach lip was increased up to 7 cm, and the riverine beach lip was higher than the side wall. For the main channel, after working condition 1 (clear water) and working condition 2 (small sand content), the elevation of the deep point of the main channel basically remained at about −8 cm, and then gradually increased by siltation to −0.8 cm. After working condition 7, the siltation elevation of the main channel was very clear. The position of the main channel remained basically unchanged; however, the position of the deep point had shifted to the right (Figure 6).

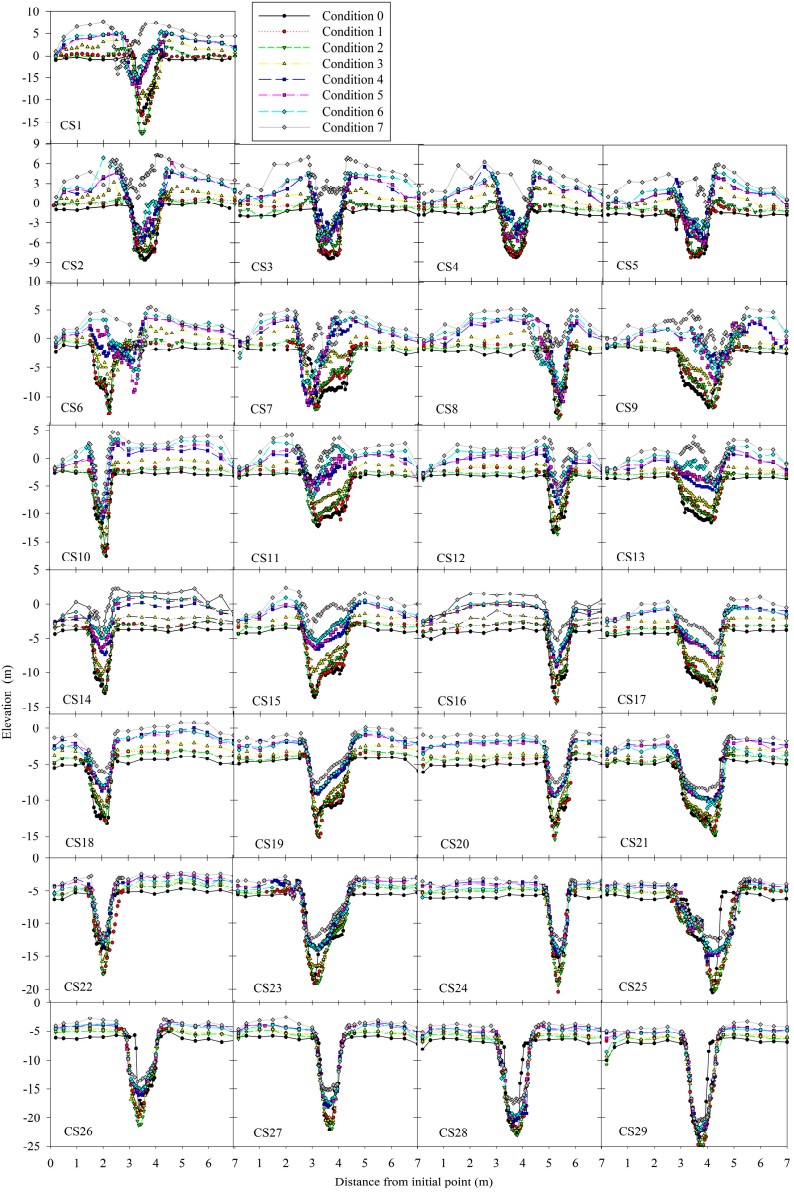

**Figure 6.** Topographic changes of each section: CS1–CS5, front riverine beach without vegetation and a straight compound channel section; CS6–CS9, with no vegetation and a meandering compound channel area on the riverine beach; CS10–CS13, the convex bank of the riverine beach with vegetation and a meandering compound channel area; CS14–CS17, the riverine beach concave bank with vegetation and a meandering compound channel area; CS18–21, with vegetation and a meandering compound channel area on both sides of the riverine beach; CS22–CS25, the flat land covered with vegetation and meandering compound channel areas; CS26–CS29, after the riverine beach without vegetation and a straight compound channel area.

(2) Adjustment of the cross-section shape of the meandering compound channel area without vegetation on riverine beach land (CS6–CS9). In section 7, the riverine beach was gradually silted up and heightened with the progress of the test, and the heightening of the riverine beach lip was greater than that of the side wall; the thalweg position of the main channel was basically unchanged on the left side of the main channel, and the width of the main channel was somewhat narrowed when the main channel was gradually silted up and heightened (Figure 6).

(3) Adjustment of the cross-section shape of the meandering compound channel area with vegetation on the convex bank of the riverine beach (CS10–CS13). In section 12, the riverine beach was gradually increased with the test, the riverine beach height was up to 6.7 cm, and the riverine beach lip height was higher than the side wall. In the main channel, the thalweg elevation of the main channel was basically unchanged at about −13 cm after condition 1 (clear water) and condition 2 (small sediment content) and then gradually increased to −6.3 cm. As the experiment continued, the main slot narrowed gradually, and the position of the main slot remained unchanged (Figure 6).

(4) Adjustment of the cross-section shape of the meandering compound channel area with vegetation on the concave bank of the riverine beach (CS14–CS17). In section 17, the riverine beach was gradually increased with the test, the riverine beach height was up to 4.7 cm, and the riverine beach lip height was higher than the side wall. The main channel was deep after working condition 1 (clear water) and working condition 2 (small sediment content), and the thalweg elevation was from −12.1 to −14.4 cm. Then the main channel was gradually increased, and the thalweg elevation was increased to −5.7 cm. The position of the main slot remained unchanged (Figure 6).

(5) Adjustment of the cross-section shape of the meandering compound channel area with vegetation on both sides of the riverine beach (CS18–CS21). In section 19, the riverine beach was gradually silted up and raised with the test, the riverine beach height was up to 4.0 cm, and the riverine beach lip height was higher than the side wall. The main channel was scoured deep after condition 1 (clear water), and the thalweg elevation was scoured deep from −13.6 to −15.2 cm. Then the main channel was gradually silted up and raised, and the thalweg elevation was increased to −7.5 cm. The position of the main slot remained unchanged (Figure 6).

(6) Adjustment of the section shape of the straight compound river channel area without vegetation on the back riverine beach (CS26–CS29). In section 24, the riverine beach was gradually silted up and heightened with the test, and the riverine beach was heightened to 3 cm. After working condition 1 (clear water), the main channel was deep, and the thalweg elevation was from −16.7 to −20.4 cm, which then gradually silted up and increased to −12.3 cm. The position of the main slot remained unchanged (Figure 6).

(7) There was no vegetation in the back floodplain, and the cross section shape of the straight compound channel area was adjusted (CS26–CS29). The riverine beach height was gradually increased with the test, and the riverine beach was increased to 3.5 cm. The thalweg elevation of the main channel was basically unchanged at about −22 cm after working conditions 1–3 and then gradually increased to −15.2 cm. The position and width of the main slot remained unchanged (Figure 6).

For the changes of velocity along the way, the transverse distribution of velocity in the upper section changed greatly, and the velocity in the lower section was relatively uniform. As the deposition of riverine beach land was mainly concentrated in the riverine beach lip, the velocity of the middle and upper section was not only concentrated in the main channel but also appeared near the left and right bank walls, and even the velocity of the side wall was greater than that of the main channel. The effect of vegetation on the flow velocity was very obvious. The vegetation on both sides of the channel, whether it was a row or full vegetation, had an effect on the stability of the main channel. When there was no vegetation on the riverine beach, the main channel was easy to move. This was able to reduce the phenomenon of excessive velocity on the riverine beach wall and increase the velocity in the main channel.

The terrain evolution law of each section was as follows. For the change of the upper and lower section, the change of the upper section was greater than that of the lower section. On the whole, the channel was silted, and the silting thickness of the upper section was greater than that of the lower section. The upper part was mainly concentrated in the riverine beach lip, and the lower part was relatively uniform.

The floodplain was gradually silted up and heightened with the progress of the test, while the main channel was deeply flushed after working condition 1 (clear water) and working condition 2 (small sediment content) and then was basically gradually silted up and heightened. After working condition 7 (coarse sediment, high sediment content), the sedimentation of the main channel was clearly higher than with the other working conditions. There was no vegetation in the upper part of the floodplain, and the main channel may move in the meandering compound channel.

The change of sediment concentration basically decreased along the way, resulting in the deposition thickness in the upper section being greater than that in the lower section. In addition, under the condition of flooding in the floodplain, the velocity of the river near the bend was generally smaller on the concave bank compared with on the convex bank, which led to the scouring of the main channel to the convex bank. This also had a trend of bending and straightening, which was consistent with the velocity distribution of the floodplain flood test (fixed bed) of the meandering compound channel with the riverine beach vegetation. However, many flood processes on the Yellow River included concave bank scouring and convex bank silting, and the velocity of the concave bank was greater than that of the convex bank [32–35]. Therefore, the suitable conditions for this experiment were a flood of a large floodplain, a flood with a large floodplain area, and a straight river channel.

## 4. Conclusions

Abundant data were obtained in this study. The results of the model test, including the transverse velocity distribution map and the topographic changes, were analyzed. With the experimental conditions in turn, the velocity distribution and cross-section topography were constantly changed, which provided a reference for the study of the erosion and deposition evolution of tidal flat vegetation on a meandering compound channel.

As the deposition of the riverine beach land was mainly concentrated at the lip of the riverine beach, the main stream was not only concentrated in the main channel but also appeared near the foot of the left and right bank levees forming flood discharge along the levees. The vegetation on the riverine beach land had a very obvious effect of slowing down the flow velocity. The vegetation on both banks, whether it was a row or full vegetation, had an effect on the stability of the main channel. The main channel was easy to move when there was no vegetation on the riverine beach land. The arrangement of full vegetation on the riverine beach had a uniform effect on the deposition and velocity distribution of the riverine beach, which could reduce excessive velocity at the foot of the riverine beach embankment and increase the velocity in the main channel.

This study was conducted under the condition of floodplain flooding, and the velocity of the river near the bend was generally smaller on the concave bank compared with on the convex bank, which led to the scouring of the main channel to the convex bank. This also had the trend of bending and straightening, which was consistent with the velocity distribution of the floodplain flooding test (fixed bed) of the meandering compound channel with riverine beach vegetation. However, many flood processes on the Yellow River were concave bank scouring and convex bank silting, and the velocity of the concave bank was greater than that of the convex bank. Therefore, the suitable conditions for this experiment included the flooding of a large floodplain, flooding with a large floodplain area, and a straight river channel.

**Author Contributions:** Conceptualization, J.L.; methodology, L.P.; validation, M.Z.; formal analysis, E.J.; investigation, A.W.; resources, Y.W.; data curation, L.P.; writing—original draft preparation, J.L.; writing—review and editing, S.J.; supervision, E.J.; project administration, S.J.; funding acquisition, J.L. All authors have read and agreed to the published version of the manuscript.

**Funding:** This research was funded by [National Natural Scientific Foundation of China], grant number [42041006, 51809106], [Natural Science Foundation of Henan Province, China], grant number [212300410200].

**Institutional Review Board Statement:** Not applicable.

**Informed Consent Statement:** Not applicable.

**Data Availability Statement:** The data presented in this study is available on request from the corresponding author.

**Acknowledgments:** We would like to thank the potential reviewer very much for their valuable comments and suggestions. We also thank my other colleagues' valuable comments and suggestions that have helped improve the manuscript. We thank MDPI (www.mdpi.com/journal/water) for its English editing during the preparation of this manuscript.

**Conflicts of Interest:** The authors declare no conflict of interest.

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
