# Peer review of "Influence of Floodplain Flooding on Channel Siltation Adjustment under the Effect of Vegetation on a Meandering Riverine Beach"

_water, doi:10.3390/w13101402_

Round 1
Reviewer 1 Report
1) Summary: what is curved vegetation? L20
2) What is "suspended secondary river" L48?
3) Line 70, you can consult the article Journal of Hydraulic Research Vol. 46, no. 5 (2008), pp. 579–597
doi: 10.3826 / jhr.2008.2986 © 2008 International Association for Hydraulic Engineering and Research "A new integrated hydromechanical model applied to flexible vegetation in riverbeds" L70
4) In the test setup, it is of interest to enter the sediment size distribution used
5) The connection between the inlet channel and the meandering channel looks like a very abrupt change. The behavior of the cross section CS6, CS8 is induced by this abrupt entry.
It is necessary to define the variable Gamma in uppercase,
6) Figure 4 shows the experimental results of the 5 different conditions. But it only shows a final profile of the terrain. For what condition is this profile.
7) The new channel formed next to the wall, is due to the wall being a hard end. The flow forms a horizontal vortex with an axis in the direction of the flow. This vortex is formed by the interaction of the flow and the rigid wall. If the wall is made of loose gravel, perhaps a weak vortex will form, or perhaps none at all.
8) figure 4 compares different section with different conditions, it is not clear if the whole channel was plenty of vegetagion along the channel or only in a meander in the setion CS22 CS24.
9) for example, Is difficult to compare the results of section Cs22 cs24 with the results of Cs8 cs10. Due to the precense ofboundary conditions. Then I dont know what happen if i compare all the results but in the middle of the channel. CS12 cs14. Is up to you.
Author Response
1)Summary: what is curved vegetation? L20
A: “curved vegetation” means “There is vegetation in the bend of the channel”.
2) What is "suspended secondary river" L48?
A: We had added the sentences as follow in L48: “Suspended secondary river is relative to the first class suspended river of the Yellow River. The bottom of the main channel is higher than the middle bank.”
3) Line 70, you can consult the article Journal of Hydraulic Research Vol. 46, no. 5 (2008), pp. 579–597 doi: 10.3826 / jhr.2008.2986 © 2008 International Association for Hydraulic Engineering and Research "A new integrated hydromechanical model applied to flexible vegetation in riverbeds" L70
A: We had added the sentence in the manuscript. Velasco, D.; Bateman, A.; Medina, V. A new integrated, hydro-mechanical model applied to flexible vegetation in riverbeds. J. Hydraul. Res. 2008, 46, 579–597.
4) In the test setup, it is of interest to enter the sediment size distribution used
A: We had added sentence in section 2.1.2 as follow: The median particle size is 0.3 mm. The particle size distribution presented a single peak, and the sediment uniformity was relatively high. D03, D06, D10, D16, D25, D50, D75, D84, D90 and D97 were 0.153 mm, 0.170 mm, 0.186 mm, 0.205 mm, 0.229 mm, 0.292 mm, 0.378 mm, 0.427 mm, 0.473 mm and 0.573 mm, respectively.
5) The connection between the inlet channel and the meandering channel looks like a very abrupt change. The behavior of the cross section CS6, CS8 is induced by this abrupt entry. It is necessary to define the variable Gamma in uppercase,
A: A very abrupt change does exist at the connection between the inlet channel and the meandering channel, which influences the behavior of the cross section CS6, CS8. But the influences are limited. We had added the sentences in section 2.1.4 as follow: The connection between the inlet channel and the meandering channel is a very abrupt change, which influences the behavior of the cross section CS6, CS8. But the in-fluences are limited, This is a transition zone, so we did not set up vegetation in the cross section CS6, CS8.
The variable Gamma in uppercase is the quadratic flow term.
6) Figure 4 shows the experimental results of the 5 different conditions. But it only shows a final profile of the terrain. For what condition is this profile.
A: The blue line is the terrain at the end of all working conditions, In other words, you can think of it as working condition 7. We had added the sentence in the Figure 4 caption.
7) The new channel formed next to the wall, is due to the wall being a hard end. The flow forms a horizontal vortex with an axis in the direction of the flow. This vortex is formed by the interaction of the flow and the rigid wall. If the wall is made of loose gravel, perhaps a weak vortex will form, or perhaps none at all.
A: Thanks for your questions, we would consider this situation in our future work.
8) figure 4 compares different section with different conditions, it is not clear if the whole channel was plenty of vegetagion along the channel or only in a meander in the setion CS22 CS24.
A: We had added (CS22 and CS24) in section 3.1-(5). CS22 and CS24 are full of vegetation on the beach.
9) for example, Is difficult to compare the results of section Cs22 cs24 with the results of Cs8 cs10. Due to the precense of boundary conditions. Then I dont know what happen if i compare all the results but in the middle of the channel. CS12 cs14. Is up to you.
A: Thanks for your suggestions. It is very difficult to compare the working conditions. Each section has its particularity, so it is inappropriate to use any section as a benchmark for comparison. Therefore, we summarize the change rule of each working condition and each section.

Reviewer 2 Report
Using both experimental simulation and theoretical approach, the authors tried to relate a compound channel model to understand the influence of beach vegetation on the suspended sediment transport and morphological evolution. In my opinion, despite the interesting theme and its merits and interest to readers, the paper has several issues that should be corrected/improved to be accepted.
A list of these issues is presented herein (in a non-specific order).
1 – The sediment transport in a floodplain is affected by much more aspects than the beach vegetation and the shape of the curved compound channel. Authors may refer to studies on the influence of vegetation, meandering or non-prismatic floodplains to find a more complete list of these aspects. The following list of references may be important for this review:
- Vojoudi Mehrabani et al. (2020) Turbulent flow structure in a vegetated non-prismatic compound channel River Research and Applications, Vol. 36(9), 1868-1878
- Pan et al. (2019) Velocity distribution characteristics in meandering compound channels with one-sided vegetated floodplains, Journal of Hydrology, Vol. 578
- Fernandes et al. (2018) Influence of floodplain and riparian vegetation in the conveyance and structure of turbulent flow in compound channels E3S Web of Conferences Volume 40, 9th International Conference on Fluvial Hydraulics, River Flow 2018
- Proust et al. (2011) Effect of different inlet flow conditions on turbulence in a straight compound open channel. 34th IAHR Congress 2011 - Balance and Uncertainty: Water in a Changing World
2 – The term used by the authors in the title and throughout the manuscript “curved beach” is somehow confusing. Firstly, authors may add the term riverine beach because “beach” is mainly used for coastal areas and can be puzzling for readers. Secondly, when authors refer to “curved” it may be better to replace this term by meandering. Is it the same? Please confirm.
3 – (line 71) There are two very different aspects expressed in this sentence. Resistance due to roughness is a completely different phenomenon than the blocking effects due to the presence of vegetation. Authors may refer to the following study:
- Brito et al. (2016) Porous media approach for RANS simulation of compound open-channel flows with submerged vegetated floodplains. Environmental Fluid Mechanics, Vol.16(6), 1247-1266
That analyses the use of porous media concept to characterize the flow in a compound channel with vegetated floodplains.
4 – All over the paper there is a confusing aspect dealing with the theoretical component and the pretical application in the Yellow river. I recommend to present all aspects as theoretical on the fundamental research and then use the Yellow river as a case study.
5 – The control of the levels by the downstream tailgate induces that the flow is assumed to be subcritical. Please refer to this aspect in the text and present the Froude and Reynolds numbers of the experiments.
6 – The quality of the figures 1 and 2 should be improved. In my opinion, authors should include a figure showing the arrangements for the five cases (for insance a schematic top view) in figure 1.
7 – Section 2.1.4 should be improved. What was measured/validated in each survey?
8 – SKM was developed for straight compound channels. How was it adapted for meandering channels?
9 – Nothing is said about the morphological evolution over time. For this experiment time to reach equilibrium is expected to be quite large. Please refer this issue in the next version.
10 – It seems that the results can be influenced by the order of the cross sections (or eventually by the time). How was this avoided in the experiments? Were the results equal if the order of the cross sections were different?
11 – Figure 6 has very low quality and should be improved.
12 – Is it possible to publish the experimental results? It would be quite useful for calibration and validation of numerical simulations.
13 – How does the sinuosity chosen for the curved compound channel compared with what is found in nature?
14 – All literature references were published more than 10 years ago and most of them more than 20 years ago. Despite its interest in a global sense, authors should update the state of the art review to more recent references. The papers referred before may be included.
15 – Line 120 – The width of the channel is repeated.
Author Response
Using both experimental simulation and theoretical approach, the authors tried to relate a compound channel model to understand the influence of beach vegetation on the suspended sediment transport and morphological evolution. In my opinion, despite the interesting theme and its merits and interest to readers, the paper has several issues that should be corrected/improved to be accepted.
A list of these issues is presented herein (in a non-specific order).
- The sediment transport in a floodplain is affected by much more aspects than the beach vegetation and the shape of the curved compound channel. Authors may refer to studies on the influence of vegetation, meandering or non-prismatic floodplains to find a more complete list of these aspects. The following list of references may be important for this review:
Vojoudi Mehrabani et al. (2020) Turbulent flow structure in a vegetated non-prismatic compound channel River Research and Applications, Vol. 36(9), 1868-1878
Pan et al. (2019) Velocity distribution characteristics in meandering compound channels with one-sided vegetated floodplains, Journal of Hydrology, Vol. 578
Fernandes et al. (2018) Influence of floodplain and riparian vegetation in the conveyance and structure of turbulent flow in compound channels E3S Web of Conferences Volume 40, 9th International Conference on Fluvial Hydraulics, River Flow 2018
Proust et al. (2011) Effect of different inlet flow conditions on turbulence in a straight compound open channel. 34th IAHR Congress 2011 - Balance and Uncertainty: Water in a Changing World
A: Thanks for your suggestions, we had read and added the references in the manuscript.
- The term used by the authors in the title and throughout the manuscript “curved beach” is somehow confusing. Firstly, authors may add the term riverine beach because “beach” is mainly used for coastal areas and can be puzzling for readers. Secondly, when authors refer to “curved” it may be better to replace this term by meandering. Is it the same? Please confirm.
A: Thanks for your suggestions. We had revised the manuscript according to the comments.
- (line 71) There are two very different aspects expressed in this sentence. Resistance due to roughness is a completely different phenomenon than the blocking effects due to the presence of vegetation. Authors may refer to the following study:
Brito et al. (2016) Porous media approach for RANS simulation of compound open-channel flows with submerged vegetated floodplains. Environmental Fluid Mechanics, Vol.16(6), 1247-1266
That analyses the use of porous media concept to characterize the flow in a compound channel with vegetated floodplains.
A: Thank you very much. We had added the reference in the manuscript.
- All over the paper there is a confusing aspect dealing with the theoretical component and the pretical application in the Yellow river. I recommend to present all aspects as theoretical on the fundamental research and then use the Yellow river as a case study.
A: Thanks for your suggestions.We conducted model tests on secondary suspended rivers of the Yellow River to preliminarily explore the effect of vegetation on river sedimentation.
5) The control of the levels by the downstream tailgate induces that the flow is assumed to be subcritical. Please refer to this aspect in the text and present the Froude and Reynolds numbers of the experiments.
A: We had added the sentences in section 2.1.5 as follow: The Reynolds number in CS24 near downstream tailgate is about 5388, and the Froude is about 0.051. The numbers in other sections are greater than these numbers.
6) The quality of the figures 1 and 2 should be improved. In my opinion, authors should include a figure showing the arrangements for the five cases (for insance a schematic top view) in figure 1.
A: We had improved the quality of the figures 1 and 2.
Figure 1. Layout of the generalized flume model.
Figure 2. Cross section of the arc top section of the meandering compound channel.
- Section 2.1.4 should be improved. What was measured/validated in each survey?
A: We had revised Section 2.1.4 as follow:
The topographic survey sections: CS1-CS29. At the end of each condition test, the shape of each section was measured.
The flow velocity measurement sections: CS6, CS8, CS10, CS12, CS14, CS16, CS18, CS20, CS22, CS24; the measuring point layout of the flow velocity measurement section: one measuring point every 0.2 m. During the test process of each condition, the velocity of the above section was monitored in real time.
The sand content measurement sections: CS3, CS8, CS12, CS16, CS20, CS24, CS27. During the test process of each condition, the sand content measurement of the above section was monitored in real time (Figure 1).
- SKM was developed for straight compound channels. How was it adapted for meandering channels?
A: We had added the sentences in section 2.2: Although SKM was developed for straight compound channels, it has been used for meandering compound channels and there have been some relevant studies [15, 31, 32].
The references as follow:
- Ervine, D.A.; Babaeyan-Koopaei, K.; Sellin, R.H.J. Two-dimensional solution for
straight and meandering overbank flows. J. Hydraul. Eng. 2000, 126, 653-669.
- Liu,C.; Wright, N.; Liu, X; Yang, K. An analytical model for lateral depth-averaged velocity distributions along a meander in curved compound channels. Adv. Water Resour. 2014, 74, 26-43.
- Shan,Y.; Liu, C.; Luo, Simple analytical model for depth-averaged velocity in meandering compound channels. Adv. Appl. Math. Mech. 2015, 36, 707-718.
- Nothing is said about the morphological evolution over time. For this experiment time to reach equilibrium is expected to be quite large. Please refer this issue in the next version.
A: We had added the sentences in section 2.15: The tests were carried out in working conditions 0 to 7 successively, and the section was measured after the end of the tests in each working condition. Restricted by the test conditions, the test duration of each working condition was 1.5h. The equilibrium is dif-ficult to achieve because the terrain is constantly changing, even at condition 7.
10) It seems that the results can be influenced by the order of the cross sections (or eventually by the time). How was this avoided in the experiments? Were the results equal if the order of the cross sections were different?
A: At the beginning of the design of the experiment, we were thinking about increasing vegetation from upstream to downstream. It is true that the results can be influenced by the order of the cross sections. The results will not equal if the order of the cross sections were different. But the laws of section velocity distributions will not change, and The main conclusions are basically unaffected.
We had added the sentences in section 2.1.3 as follow: Based on practical considerations, vegetation increased gradually from upstream to downstream.Therefore, our model gradually increases vegetation from upstream to downstream (from CS1-CS29).
- Figure 6 has very low quality and should be improved.
A: We had improved the quality of Figure 6 as follow:
- Is it possible to publish the experimental results? It would be quite useful for calibration and validation of numerical simulations.
A: We had upload part of data in the website of the journal, we hope it would help more researchers.
- How does the sinuosity chosen for the curved compound channel compared with what is found in nature?
A: Based on the practical consideration of the form of meandering river in the lower reaches of the Yellow River, we established a meandering compound channel model. The following figure shows the channel form map of the lower reaches of the Yellow River.
Fig. the lower reaches of the Yellow River
We had added the sentences in section 2.1.1 as follow: Based on the practical consideration of the form of meandering river in the lower reaches of the Yellow River, we established a meandering compound channel model.
- All literature references were published more than 10 years ago and most of them more than 20 years ago. Despite its interest in a global sense, authors should update the state of the art review to more recent references. The papers referred before may be included.
A: Thanks for your suggestions, we had updated some references.
- Line 120 – The width of the channel is repeated.
A: We had deleted the sentence.
Using both experimental simulation and theoretical approach, the authors tried to relate a compound channel model to understand the influence of beach vegetation on the suspended sediment transport and morphological evolution. In my opinion, despite the interesting theme and its merits and interest to readers, the paper has several issues that should be corrected/improved to be accepted.
A list of these issues is presented herein (in a non-specific order).
- The sediment transport in a floodplain is affected by much more aspects than the beach vegetation and the shape of the curved compound channel. Authors may refer to studies on the influence of vegetation, meandering or non-prismatic floodplains to find a more complete list of these aspects. The following list of references may be important for this review:
Vojoudi Mehrabani et al. (2020) Turbulent flow structure in a vegetated non-prismatic compound channel River Research and Applications, Vol. 36(9), 1868-1878
Pan et al. (2019) Velocity distribution characteristics in meandering compound channels with one-sided vegetated floodplains, Journal of Hydrology, Vol. 578
Fernandes et al. (2018) Influence of floodplain and riparian vegetation in the conveyance and structure of turbulent flow in compound channels E3S Web of Conferences Volume 40, 9th International Conference on Fluvial Hydraulics, River Flow 2018
Proust et al. (2011) Effect of different inlet flow conditions on turbulence in a straight compound open channel. 34th IAHR Congress 2011 - Balance and Uncertainty: Water in a Changing World
A: Thanks for your suggestions, we had read and added the references in the manuscript.
- The term used by the authors in the title and throughout the manuscript “curved beach” is somehow confusing. Firstly, authors may add the term riverine beach because “beach” is mainly used for coastal areas and can be puzzling for readers. Secondly, when authors refer to “curved” it may be better to replace this term by meandering. Is it the same? Please confirm.
A: Thanks for your suggestions. We had revised the manuscript according to the comments.
- (line 71) There are two very different aspects expressed in this sentence. Resistance due to roughness is a completely different phenomenon than the blocking effects due to the presence of vegetation. Authors may refer to the following study:
Brito et al. (2016) Porous media approach for RANS simulation of compound open-channel flows with submerged vegetated floodplains. Environmental Fluid Mechanics, Vol.16(6), 1247-1266
That analyses the use of porous media concept to characterize the flow in a compound channel with vegetated floodplains.
A: Thank you very much. We had added the reference in the manuscript.
- All over the paper there is a confusing aspect dealing with the theoretical component and the pretical application in the Yellow river. I recommend to present all aspects as theoretical on the fundamental research and then use the Yellow river as a case study.
A: Thanks for your suggestions.We conducted model tests on secondary suspended rivers of the Yellow River to preliminarily explore the effect of vegetation on river sedimentation.
5) The control of the levels by the downstream tailgate induces that the flow is assumed to be subcritical. Please refer to this aspect in the text and present the Froude and Reynolds numbers of the experiments.
A: We had added the sentences in section 2.1.5 as follow: The Reynolds number in CS24 near downstream tailgate is about 5388, and the Froude is about 0.051. The numbers in other sections are greater than these numbers.
6) The quality of the figures 1 and 2 should be improved. In my opinion, authors should include a figure showing the arrangements for the five cases (for insance a schematic top view) in figure 1.
A: We had improved the quality of the figures 1 and 2.
Figure 1. Layout of the generalized flume model.
Figure 2. Cross section of the arc top section of the meandering compound channel.
- Section 2.1.4 should be improved. What was measured/validated in each survey?
A: We had revised Section 2.1.4 as follow:
The topographic survey sections: CS1-CS29. At the end of each condition test, the shape of each section was measured.
The flow velocity measurement sections: CS6, CS8, CS10, CS12, CS14, CS16, CS18, CS20, CS22, CS24; the measuring point layout of the flow velocity measurement section: one measuring point every 0.2 m. During the test process of each condition, the velocity of the above section was monitored in real time.
The sand content measurement sections: CS3, CS8, CS12, CS16, CS20, CS24, CS27. During the test process of each condition, the sand content measurement of the above section was monitored in real time (Figure 1).
- SKM was developed for straight compound channels. How was it adapted for meandering channels?
A: We had added the sentences in section 2.2: Although SKM was developed for straight compound channels, it has been used for meandering compound channels and there have been some relevant studies [15, 31, 32].
The references as follow:
- Ervine, D.A.; Babaeyan-Koopaei, K.; Sellin, R.H.J. Two-dimensional solution for
straight and meandering overbank flows. J. Hydraul. Eng. 2000, 126, 653-669.
- Liu,C.; Wright, N.; Liu, X; Yang, K. An analytical model for lateral depth-averaged velocity distributions along a meander in curved compound channels. Adv. Water Resour. 2014, 74, 26-43.
- Shan,Y.; Liu, C.; Luo, Simple analytical model for depth-averaged velocity in meandering compound channels. Adv. Appl. Math. Mech. 2015, 36, 707-718.
- Nothing is said about the morphological evolution over time. For this experiment time to reach equilibrium is expected to be quite large. Please refer this issue in the next version.
A: We had added the sentences in section 2.15: The tests were carried out in working conditions 0 to 7 successively, and the section was measured after the end of the tests in each working condition. Restricted by the test conditions, the test duration of each working condition was 1.5h. The equilibrium is dif-ficult to achieve because the terrain is constantly changing, even at condition 7.
10) It seems that the results can be influenced by the order of the cross sections (or eventually by the time). How was this avoided in the experiments? Were the results equal if the order of the cross sections were different?
A: At the beginning of the design of the experiment, we were thinking about increasing vegetation from upstream to downstream. It is true that the results can be influenced by the order of the cross sections. The results will not equal if the order of the cross sections were different. But the laws of section velocity distributions will not change, and The main conclusions are basically unaffected.
We had added the sentences in section 2.1.3 as follow: Based on practical considerations, vegetation increased gradually from upstream to downstream.Therefore, our model gradually increases vegetation from upstream to downstream (from CS1-CS29).
- Figure 6 has very low quality and should be improved.
A: We had improved the quality of Figure 6 as follow:
- Is it possible to publish the experimental results? It would be quite useful for calibration and validation of numerical simulations.
A: We had upload part of data in the website of the journal, we hope it would help more researchers.
- How does the sinuosity chosen for the curved compound channel compared with what is found in nature?
A: Based on the practical consideration of the form of meandering river in the lower reaches of the Yellow River, we established a meandering compound channel model. The following figure shows the channel form map of the lower reaches of the Yellow River.
Fig. the lower reaches of the Yellow River
We had added the sentences in section 2.1.1 as follow: Based on the practical consideration of the form of meandering river in the lower reaches of the Yellow River, we established a meandering compound channel model.
- All literature references were published more than 10 years ago and most of them more than 20 years ago. Despite its interest in a global sense, authors should update the state of the art review to more recent references. The papers referred before may be included.
A: Thanks for your suggestions, we had updated some references.
- Line 120 – The width of the channel is repeated.
A: We had deleted the sentence.

Reviewer 3 Report
Very nice paper, please correct the minor points in the attached PDF.

Author Response
Very nice paper, please correct the minor points in the attached PDF.
1)how you support these lenghts?
A: We had revised the sentence as follow: Disposable chopsticks (22.5 cm in length and 5 mm in diameter) were glued on the riverine beach to simulate vegetation.
2)improve figure resolution in the lower panel.
A: We had revised the figure 1, 2 and 6.
3)Please check and add the following papers:
Quesada-Román, A., Ballesteros-Cánovas, J.A., Granados-Bolaños, S., Birkel, C., & Stoffel, M. (2020). Dendrogeomorphic reconstruction of floods in a dynamic tropical river. Geomorphology, 359, 107133. https://doi.org/10.1016/j.geomorph.2020.107133
Quesada-Román, A., & Villalobos-Chacón, A. (2020). Flash flood impacts of Hurricane Otto and hydrometeorological risk mapping in Costa Rica. Geografisk Tidsskrift-Danish Journal of Geography, 120(2), 142-155. https://doi.org/10.1080/00167223.2020.1822195
You can see the effects of vegetation during intense extreme rainfall events in tropical conditions. Therefore, you can use your modelling results and discuss with papers as the ones indicated.
A: We had added the two references in the manuscript.
4) improve figure resolution
A: We had revised the figure 6 and improve the resolution.
